# Message Passing Inference for Large Scale Graphical Models with High Order Potentials

**Jian Zhang**
ETH Zurich
jizhang@ethz.ch

**Alexander G. Schwing**
University of Toronto
aschwing@cs.toronto.edu

**Raquel Urtasun**
University of Toronto
urtasun@cs.toronto.edu

## Abstract

To keep up with the Big Data challenge, parallelized algorithms based on dual decomposition have been proposed to perform inference in Markov random fields. Despite this parallelization, current algorithms struggle when the energy has high order terms and the graph is densely connected. In this paper we propose a partitioning strategy followed by a message passing algorithm which is able to exploit pre-computations. It only updates the high-order factors when passing messages across machines. We demonstrate the effectiveness of our approach on the task of joint layout and semantic segmentation estimation from single images, and show that our approach is orders of magnitude faster than current methods.

## 1 Introduction

Graphical models are a very useful tool to capture the dependencies between the variables of interest. In domains such as computer vision, natural language processing and computational biology they have been very widely used to solve problems such as semantic segmentation [37], depth reconstruction [21], dependency parsing [4, 25] and protein folding [36].

Despite decades of research, finding the maximum a-posteriori (MAP) assignment or the minimimum energy configuration remains an open problem, as it is NP-hard in general. Notable exceptions are specialized solvers such as graph-cuts [7, 3] and dynamic programming [19, 1], which retrieve the global optima for sub-modular energies and tree-shaped graphs. Algorithms based on message passing [18, 9], a series of graph cut moves [16] or branch-and-bound techniques [5] are common choices to perform approximate inference in the more general case. A task closely related to MAP inference but typically harder is computation of the probability for a given configuration. It requires computing the partition function, which is typically done via message passing [18], sampling or by repeatedly using MAP inference to solve tasks perturbed via Gumbel distributions [8].

Of particular difficulty is the case where the involved potentials depend on more than two variables, *i.e.*, they are high-order, or the graph is densely connected. Several techniques have been developed to allow current algorithms to handle high-order potentials, but they are typically restricted to potentials of a specific form, *e.g.*, a function of the cardinality [17] or piece-wise linear potentials [11, 10]. For densely connected graphs with Gaussian potentials efficient inference methods based on filtering have been proposed [14, 33].

Alternating minimization approaches, which iterate between solving for subsets of variables have also been studied [32, 38, 29]. However, most approaches loose their guarantees since related subproblems are solved independently. Another method to improve computational efficiency is to divide the model into smaller tasks, which are solved in parallel using dual decomposition techniques [13, 20, 22]. Contrasting alternating minimization, convergence properties are ensured. However, these techniques are computationally expensive despite the division of computation, since global and dense interactions are still present.

In this work we show that for many graphical models it is possible to devise a partitioning strategy followed by a message passing algorithm such that efficiency can be improved significantly. In particular, our approach adds additional terms to the energy function (*i.e.*, regions to the Hasse diagram) such that the high-order factors can be pre-computed and remain constant during local message passing within each machine. As a consequence, high-order factors are only accessed once before sending messages across machines. This contrasts tightening approaches [27, 28, 2, 26], where additional regions are added to better approximate the marginal polytope at the cost of additional computations, while we are mainly interested in computational efficiency. In contrast to re-scheduling strategies [6, 30, 2], our rescheduling is fixed and does not require additional computation.

Our experimental evaluations show that state-of-the-art techniques [9, 22] have difficulties optimizing energy functions that correspond to densely connected graphs with high-order factors. In contrast our approach is able to achieve more than one order of magnitude speed-ups while retrieving the same solution in the complex task of jointly estimating 3D room layout and image segmentation from a single RGB-D image.

## 2   Background: Dual Decomposition for Message Passing

We start by reviewing dual-decomposition approaches for inference in graphical models with high-order factors. To this end, we consider distributions defined over a discrete domain $\mathcal{S} = \prod_{i=1}^{N} \mathcal{S}_i$, which is composed of a product of $N$ smaller discrete spaces $\mathcal{S}_i = \{1, \ldots, |\mathcal{S}_i|\}$. We model our distribution to depend log-linearly on a scoring function $\theta(s)$ defined over the aforementioned discrete product space $\mathcal{S}$, *i.e.*, $p(s) = \frac{1}{Z} \exp \theta(s)$, with $Z$ the partition function. Given the scoring function $\theta(s)$ of a configuration $s$, it is unfortunately generally #P-complete to compute its probability since the partition function $Z$ is required. Its logarithm equals the following variational program [12]:

$$\log Z = \max_{p \in \Delta} \sum_{s} p(s)\theta(s) + H(p), \qquad (1)$$

where $H$ denotes the entropy and $\Delta$ indicates the probability simplex.

The variational program in Eq. (1) is challenging as it operates on the exponentially sized domain $\mathcal{S}$. However, we can make use of the fact that for many relevant applications the scoring function $\theta(s)$ is additively composed of local terms, *i.e.*, $\theta(s) = \sum_{r \in \mathcal{R}} \theta_r(s_r)$. These local scoring functions $\theta_r$ depend on a subset of variables $s_r = (s_i)_{i \in r}$, defined on a domain $\mathcal{S}_r \subseteq \mathcal{S}$, which is specified by the restriction often referred to as region $r \subseteq \{1, \ldots, N\}$, *i.e.*, $\mathcal{S}_r = \prod_{i \in r} \mathcal{S}_i$. We refer to $\mathcal{R}$ as the set of all restriction required to compute the scoring function $\theta$.

Locality of the scoring function allows to equivalently rewrite the expected score via $\sum_s p(s)\theta(s) = \sum_{r,s_r} p_r(s_r)\theta_r(s_r)$ by employing marginals $p_r(s_r) = \sum_{s \setminus s_r} p(s)$. Unfortunately an exact decomposition of the entropy $H(p)$ using marginals is not possible. Instead, the entropy is typically approximated by a weighted sum of local entropies $H(p) \approx \sum_r c_r H(p_r)$, with $c_r$ the counting numbers. The task remains intractable despite the entropy approximation since the marginals $p_r(s_r)$ are required to arise from a valid joint distribution $p(s)$. However, if we require the marginals to be consistent only locally, we obtain a tractable approximation [34]. We thus introduce local beliefs $b_r(s_r)$ to denote the approximation, not to be confused with the true marginals $p_r$. The beliefs are required to fulfill local marginalization constraints, *i.e.*, $\sum_{s_p \setminus s_r} b_p(s_p) = b_r(s_r) \, \forall r, s_r, p \in P(r)$, where the set $P(r)$ subsumes the set of all parents of region $r$ for which we want marginalization to hold.

Putting all this together, we obtain the following approximation:

$$\max_{b} \quad \sum_{r,s_r} b_r(s_r)\theta_r(s_r) + \sum_{r} c_r H(b_r)$$

$$\text{s.t.} \quad \forall r \quad b_r \in \mathcal{C} = \left\{ b_r : \begin{array}{l} b_r \in \Delta \\ \sum_{s_p \setminus s_r} b_p(s_p) = b_r(s_r) \quad \forall s_r, p \in P(r). \end{array} \right. \qquad (2)$$

The computation and memory requirements can be too demanding when dealing with large graphical models. To address this issue, [13, 22] showed that this task can be distributed onto multiple

**Algorithm: Distributed Message Passing Inference**
Let $a = 1/|M(r)|$ and **repeat until convergence**

1. For every $\kappa$ in parallel: **iterate $T$ times** over $r \in R(\kappa)$:

$\forall p \in P(r), s_r$

$$\mu_{p \to r}(s_r) = \epsilon \hat{c}_p \ln \sum_{s_p \setminus s_r} \exp \frac{\hat{\theta}_p(s_p) - \sum\limits_{p' \in P(p)} \lambda_{p \to p'}(s_p) + \sum\limits_{r' \in C(p) \cap \kappa \setminus r} \lambda_{r' \to p}(s_{r'}) + \nu_{\kappa \to p}(s_p)}{\epsilon \hat{c}_p} \quad (3)$$

$$\lambda_{r \to p}(s_r) \propto \frac{\hat{c}_p}{\hat{c}_r + \sum\limits_{p \in P(r)} \hat{c}_p} \left( \hat{\theta}_r(s_r) + \sum_{c \in C(r) \cap \kappa} \lambda_{c \to r}(s_c) + \nu_{\kappa \to r}(s_r) + \sum_{p \in P(r)} \mu_{p \to r}(s_r) \right) - \mu_{p \to r}(s_r) \quad (4)$$

2. Exchange information by iterating once over $r \in G \; \forall \kappa \in M(r)$

$$\nu_{\kappa \to r}(s_r) = a \sum_{c \in C(r)} \lambda_{c \to r}(s_c) - \sum_{c \in C(r) \cap \kappa} \lambda_{c \to r}(s_c) + \sum_{p \in P(r)} \lambda_{r \to p}(s_r) - a \sum_{\kappa \in M(r), p \in P(r)} \lambda_{r \to p}(s_r) \quad (5)$$

Figure 1: A block-coordinate descent algorithm for the distributed inference task.

computers $\kappa$ by employing dual decomposition techniques. More specifically, the task is partitioned into multiple independent tasks with constraints at the boundary ensuring consistency of the parts upon convergence. Hence, an additional constraint is added to make sure that all beliefs $b_r^\kappa$ that are assigned to multiple computers, *i.e.*, those at the boundary of the parts, are consistent upon convergence and equal a single region belief $b_r$. The distributed program is then:

$$\max_{b_r, b_r^\kappa \in \Delta} \quad \sum_{\kappa, r, s_r} b_r^\kappa(s_r) \hat{\theta}_r(s_r) + \sum_{\kappa, r} \hat{c}_r H(b_r^\kappa)$$

$$\text{s.t.} \quad \begin{aligned} \forall \kappa, r \in \mathcal{R}_\kappa, s_r, p \in P(r) \quad & \sum_{s_p \setminus s_r} b_p^\kappa(s_p) = b_r^\kappa(s_r) \\ \forall \kappa, r \in \mathcal{R}_\kappa, s_r \quad & b_r^\kappa(s_r) = b_r(s_r), \end{aligned}$$

where $\mathcal{R}_\kappa$ refers to regions on comptuer $\kappa$. We uniformly distributed the scores $\theta_r(s_r)$ and the counting numbers $c_r$ of a region $r$ to all overlapping machines. Thus $\hat{\theta}_r = \theta_r/|M(r)|$ and $\hat{c}_r = c_r/|M(r)|$ with $M(r)$ the set of machines that are assigned to region $r$.

Note that this program operates on the regions defined by the energy decomposition. To derive an efficient algorithm making use of the structure incorporated in the constraints we follow [22] and change to the dual domain. For the marginalization constraints we introduce Lagrange multipliers $\lambda_{r \to p}^\kappa(s_r)$ for every computer $\kappa$, all regions $r \in \mathcal{R}_\kappa$ assigned to that computer, all its states $s_r$ and all its parents $p$. For the consistency constraint we introduce Lagrange multipliers $\nu_{\kappa \to r}(s_r)$ for all computers, regions and states. The arrows indicate that the Lagrange multipliers can be interpreted as messages sent between different nodes in a Hasse diagram with nodes corresponding to the regions.

The resulting distributed inference algorithm [22] is summarized in Fig. 1. It consists of two parts, the first of which is a standard message passing on the Hasse-diagram defined locally on each computer $\kappa$. The second operation interrupts message passing occasionally to exchange information between computers. This second task of exchanging messages is often visualized on a graph $G$ with nodes corresponding to computers and additional vertices denoting shared regions.

Fig. 2(a) depicts a region graph with four unary regions and two high-order ones, *i.e.*, $\mathcal{R} = \{\{1\}, \{2\}, \{3\}, \{4\}, \{1, 2, 3\}, \{1, 2, 3, 4\}\}$. We partition this region graph onto two computers $\kappa_1, \kappa_2$ as indicated via the dashed rectangles. The graph $G$ containing as nodes both computers and the shared region is provided as well. The connections between all regions are labeled with the corresponding message, *i.e.*, $\lambda$, $\mu$ and $\nu$. We emphasize that the consistency messages $\nu$ are only modified when sending information between computers $\kappa$. Investigating the provided example in Fig. 2(a) more carefully we observe that the computation of $\mu$ as defined in Eq. (3) in Fig. 1 involves summing over the state-space of the third-order region $\{1, 2, 3\}$ and the fourth-order region $\{1, 2, 3, 4\}$. The presence of those high-order regions make dual decomposition approaches [22]

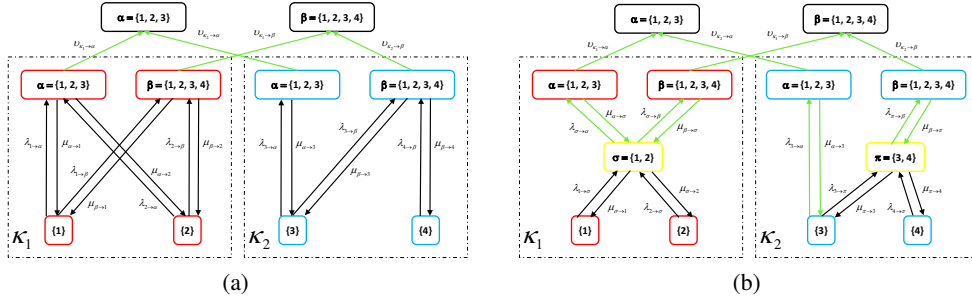

Figure 2: Standard distributed message passing operating on an inference task partitioned to two computers (left) is compared to the proposed approach (right) where newly introduced regions (yellow) ensure constant messages $\mu$ from the high-order regions.

impractical. In the next section we show how message passing algorithms can become orders of magnitude faster when adding additional regions.

## 3 Efficient Message Passing for High-order Models

The distributed message passing procedure described in the previous section involves summations over large state-spaces when computing the messages $\mu$. In this section we derive an approach that can significantly reduce the computation by adding additional regions and performing message-passing with a specific message scheduling. Our key observation is that computation can be greatly reduced if the high-order regions are singly-connected since their outgoing message $\mu$ remains constant. Generally, singly-connected high-order regions do not occur in graphical models. However, in many cases we can use dual decomposition to distribute the computation in a way that the high-order regions become singly-connected if we introduce additional intermediate regions located between the high-order regions and the low-order ones (*e.g.*, unary regions).

At first sight, adding regions increases computational complexity since we have to iterate over additional terms. However, we add regions only if they result in constant messages from regions with even larger state space. By pre-computing those constant messages rather than re-evaluating them at every iteration, we hence decrease computation time despite augmenting the graph with additional regions, *i.e.*, additional marginal beliefs $b_r$.

Specifically, we observe that there are no marginalization constraints for the singly-connected high-order regions, subsumed in the set $\mathcal{H}_\kappa = \{r \in \hat{\mathcal{R}}_\kappa : P(r) = \emptyset, |C(r)| = 1\}$, since their set of parents is empty. An important observation made precise in Claim 1 is that the corresponding messages $\mu$ are constant for high-order regions unless $\nu_{\kappa \to r}$ changes. Therefore we can improve the message passing algorithm discussed in the previous section by introducing additional regions to increase the size of the set $|\mathcal{H}_\kappa|$ as much as possible while not changing the cost function. The latter is ensured by requiring the additional counting numbers and potentials to equal zero. However, we note that the program will change since the constraint set is augmented.

More formally, let $\hat{\mathcal{R}}_\kappa$ be the set of all regions, *i.e.*, the regions $\mathcal{R}_\kappa$ of the original task on computer $\kappa$ in addition to the newly added regions $\hat{r} \in \hat{\mathcal{R}}_\kappa \setminus \mathcal{R}_\kappa$. Let $\mathcal{H}_\kappa = \{r \in \hat{\mathcal{R}}_\kappa : P(r) = \emptyset, |C(r)| = 1\}$ be the set of high-order regions on computer $\kappa$ that are singly connected and have no parent. Further, let its complement $\overline{\mathcal{H}}_\kappa = \hat{\mathcal{R}}_\kappa \setminus \mathcal{H}_\kappa$ denote all remaining regions. The inference task is given by

$$\max_{b_r, b_r^\kappa \in \Delta} \quad \sum_{\kappa, r, s_r} b_r^\kappa(s_r)\hat{\theta}_r(s_r) + \sum_{\kappa, r} \hat{c}_r H(b_r^\kappa)$$

$$\text{s.t.} \quad \begin{aligned} &\forall \kappa, r \in \overline{\mathcal{H}}_\kappa, s_r, p \in P(r) \quad \sum_{s_p \setminus s_r} b_p^\kappa(s_p) = b_r^\kappa(s_r) \\ &\forall \kappa, r \in \hat{\mathcal{R}}_\kappa, s_r \qquad\qquad\qquad b_r^\kappa(s_r) = b_r(s_r). \end{aligned} \quad (9)$$

Even though we set $\theta_r(s_r) \equiv 0$ for all states $s_r$, and $\hat{c}_r = 0$ for all newly added regions $r \in \hat{\mathcal{R}}_\kappa \setminus \mathcal{R}_\kappa$, the inference task is not identical to the original problem since the constraint set is not the same. Note that new regions introduce new marginalization constraints. Next we show that messages leaving singly-connected high-order regions are constant.

---
**Algorithm: Message Passing for Large Scale Graphical Models with High Order Potentials**
Let $a = 1/|M(r)|$ and **repeat until convergence**

1. For every $\kappa$ in parallel: Update singly-connected regions $p \in \mathcal{H}_\kappa$: let $r = C(p)$ $\forall s_r$

$$\mu_{p \to r}(s_r) = \epsilon \hat{c}_p \ln \sum_{s_p \backslash s_r} \exp \frac{\hat{\theta}_p(s_p) - \sum\limits_{p' \in P(p)} \lambda_{p \to p'}(s_p) + \sum\limits_{r' \in C(p) \cap \kappa \backslash r} \lambda_{r' \to p}(s_{r'}) + \nu_{\kappa \to p}(s_p)}{\epsilon \hat{c}_p}$$

2. For every $\kappa$ in parallel: **iterate $T$ times** over $r \in \hat{\mathcal{R}}_\kappa$:

$\forall p \in P(r) \backslash \mathcal{H}_\kappa, s_r$

$$\mu_{p \to r}(s_r) = \epsilon \hat{c}_p \ln \sum_{s_p \backslash s_r} \exp \frac{\hat{\theta}_p(s_p) - \sum\limits_{p' \in P(p)} \lambda_{p \to p'}(s_p) + \sum\limits_{r' \in C(p) \cap \kappa \backslash r} \lambda_{r' \to p}(s_{r'}) + \nu_{\kappa \to p}(s_p)}{\epsilon \hat{c}_p} \quad (6)$$

$\forall p \in P(r), s_r$

$$\lambda_{r \to p}(s_r) \propto \frac{\hat{c}_p}{\hat{c}_r + \sum\limits_{p \in P(r)} \hat{c}_p} \left( \hat{\theta}_r(s_r) + \sum_{c \in C(r) \cap \kappa} \lambda_{c \to r}(s_c) + \nu_{\kappa \to r}(s_r) + \sum_{p \in P(r)} \mu_{p \to r}(s_r) \right) - \mu_{p \to r}(s_r) \quad (7)$$

3. Exchange information by iterating once over $r \in G$ $\forall \kappa \in M(r)$

$$\nu_{\kappa \to r}(s_r) = a \sum_{c \in C(r)} \lambda_{c \to r}(s_c) - \sum_{c \in C(r) \cap \kappa} \lambda_{c \to r}(s_c) + \sum_{p \in P(r)} \lambda_{r \to p}(s_r) - a \sum_{\kappa \in M(r), p \in P(r)} \lambda_{r \to p}(s_r) \quad (8)$$

---

Figure 3: A block-coordinate descent algorithm for the distributed inference task.

**Claim 1.** *During message passing updates defined in Fig. 1 the multiplier $\mu_{p \to r}(s_r)$ is constant for singly-connected high-order regions $p$.*

**Proof:** More carefully investigating Eq. (3) which defines $\mu$, it follows that $\sum_{p' \in P(p)} \lambda_{p \to p'}(s_p) = 0$ because $P(p) = \emptyset$ since $p$ is assumed singly-connected. For the same reason we obtain $\sum_{r' \in C(p) \cap \kappa \backslash r} \lambda_{r' \to p}(s_{r'}) = 0$ because $r' \in C(p) \cap \kappa \backslash r = \emptyset$ and $\nu_{\kappa \to p}(s_p)$ is constant upon each exchange of information. Therefore, $\mu_{p \to r}(s_r)$ is constant irrespective of all other messages and can be pre-computed upon exchange of information. ∎

We can thus pre-compute the constant messages before performing message passing. Our approach is summarized in Fig. 3. We now provide its convergence properties in the following claim.

**Claim 2.** *The algorithm outlined in Fig. 3 is guaranteed to converge to the global optimum of the program given in Eq. (9) for $\epsilon c_r > 0$ $\forall r$ and is guaranteed to converge in case $\epsilon c_r \geq 0$ $\forall r$.*

**Proof:** The message passing algorithm is derived as a block-coordinate descent algorithm in the dual domain. Hence it inherits the properties of block-coordinate descent algorithms [31] which are guaranteed to converge to a single global optimum in case of strict concavity ($\epsilon c_r > 0$ $\forall r$) and which are guaranteed to converge in case of concavity only ($\epsilon c_r \geq 0$ $\forall r$), which proves the claim. ∎

We note that Claim 1 nicely illustrates the benefits of working with region graphs rather than factor graphs. A bi-partite factor graph contains variable nodes connected to possibly high-order factors. Assume that we distributed the task at hand such that every high-order region of size larger than two is connected to at most two local variables. By adding a pairwise region in between the original high-order factor node and the variable nodes we are able to reduce computational complexity since the high-order factors are now singly connected. Therefore, we can guarantee that the complexity of the local message-passing steps run in each machine reduces from the state-space size of the largest factor to the size of the largest newly introduced region in each computer. This is summarized in the following claim.

**Claim 3.** *Assume we are given a high-order factor-graph representation of a graphical model. By distributing the model onto multiple computers and by introducing additional regions we reduce the complexity of the message passing iterations on every computer generally dominated by the state-*

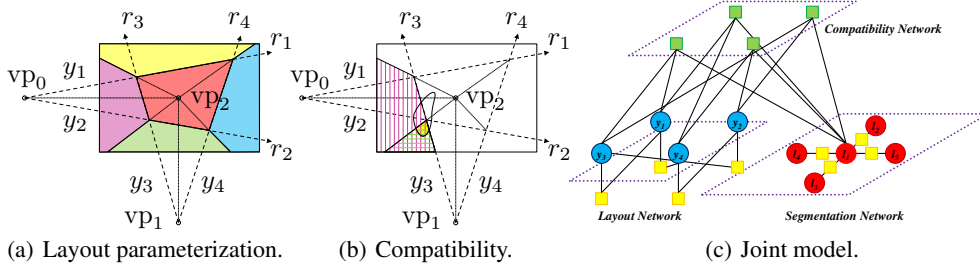

|  | (a) Layout parameterization. | (b) Compatibility. | (c) Joint model. |

Figure 4: Parameterization of the layout task is visualized in (a). Compatibility of a superpixel labeling with a wall parameterization using third-order functions is outlined in (b) and the graphical model for the joint layout-segmentation task is depicted in (c).

| rel. duality gap | 1 | 0.1 | 0.01 | rel. duality gap | 1 | 0.1 | 0.01 |
|---|---|---|---|---|---|---|---|
| Ours [s] | **0.78** | **5.92** | **51.59** | Ours [s] | **15.58** | **448.26** | **1150.1** |
| cBP [s] | 31.60 | 986.54 | 1736.6 | cBP [s] | 411.81 | 4357.9 | 4479.9 |
| dcBP [s] | 19.48 | 1042.8 | 1772.6 | dcBP [s] | 451.71 | 4506.6 | 4585.3 |
| | | $\epsilon = 0$ | | | | $\epsilon = 1$ | |

Table 1: Average time to achieve the specified relative duality gap for $\epsilon = 0$ (left) and $\epsilon = 1$ (right).

space size of the largest region $s_{\max} = \max_{r \in \mathcal{R}_\kappa} |\mathcal{S}_r|$ from $O(s_{\max})$ to $O(s'_{\max})$ with $s'_{\max} = \max_{r \in \hat{\mathcal{R}}_\kappa} |\mathcal{S}_{r \cap \overline{\mathcal{H}}_\kappa}|$.

**Proof:** The complexity of standard message passing on a region graph is linear on the largest state-space region, *i.e.*, $O(s_{\max})$. Since some operations can be pre-computed as per Claim 1 we emphasize that the largest newly introduced region on computer $\kappa$ is of state-space size $s'_{\max}$ which concludes the proof. ∎

Claim 3 indicates that distributing computation in addition to message rescheduling is a powerful tool to cope with high-order potentials. To gain some insight, we illustrate our idea with a specific example. Suppose we distribute the inference computation on two computers $\kappa_1$, $\kappa_2$ as shown in Fig. 2(a). We compare it to a task on $\hat{\mathcal{R}}$ regions, *i.e.*, we introduce additional regions $\hat{r} \in \hat{\mathcal{R}} \setminus \mathcal{R}$. The messages required in the augmented task are visualized in Fig. 2(b). Each computer (box highlighted with dashed lines) is assigned a task specified by the contained region graph. As before we also visualize the messages $\nu$ occasionally sent between the computers in a graph containing as nodes the shared factors and the computers (boxes drawn with dashed lines). The algorithm proceeds by passing messages $\lambda, \mu$ on each computer independently for $T$ rounds. Afterwards messages $\nu$ are exchanged between computers. Importantly, we note that messages for singly-connected high-order regions within dashed boxes are only required to be computed once upon exchanging message $\nu$. This is the case for all high-order regions in Fig. 2(b) and for no high-order region in Fig. 2(a), highlighting the obtained computational benefits.

## 4  Experimental Evaluation

We demonstrate the effectiveness of our approach in the task of jointly estimating the layout and semantic labels of indoor scenes from a single RGB-D image. We use the dataset of [38], which is a subset of the NYU v2 dataset [24]. Following [38], we utilize 202 images for training and 101 for testing. Given the vanishing points (points where parallel lines meet at infinity), the layout task can be formulated with four random variables $s_1, \ldots, s_4$, each of which corresponds to angles for rays originating from two distinct vanishing points [15]. We discretize each ray into $|\mathcal{S}_i| = 25$ states. To define the segmentation task, we partition each image into super pixels. We then define a random variable with six states for each super pixel $s_i \in \mathcal{S}_i = \{\text{left}, \text{front}, \text{right}, \text{ceiling}, \text{floor}, \text{clutter}\}$ with $i > 4$. We refer the reader to Fig. 4(a) and Fig. 4(b) for an illustration of the parameterization of the problem. The graphical model for the joint problem is depicted in Fig. 4(c).

The score of the joint model is given by a sum of scores

$$\theta(s) = \theta_{\text{lay}}(s_1, \ldots, s_4) + \theta_{\text{label}}(s_5, \ldots, s_{M+4}) + \theta_{\text{comp}}(s),$$

where $\theta_{\text{lay}}$ is defined as the sum of scores over the layout faces, which can be decomposed into a sum of pairwise functions using integral geometry [23]. The labeling score $\theta_{\text{label}}$ contains unary

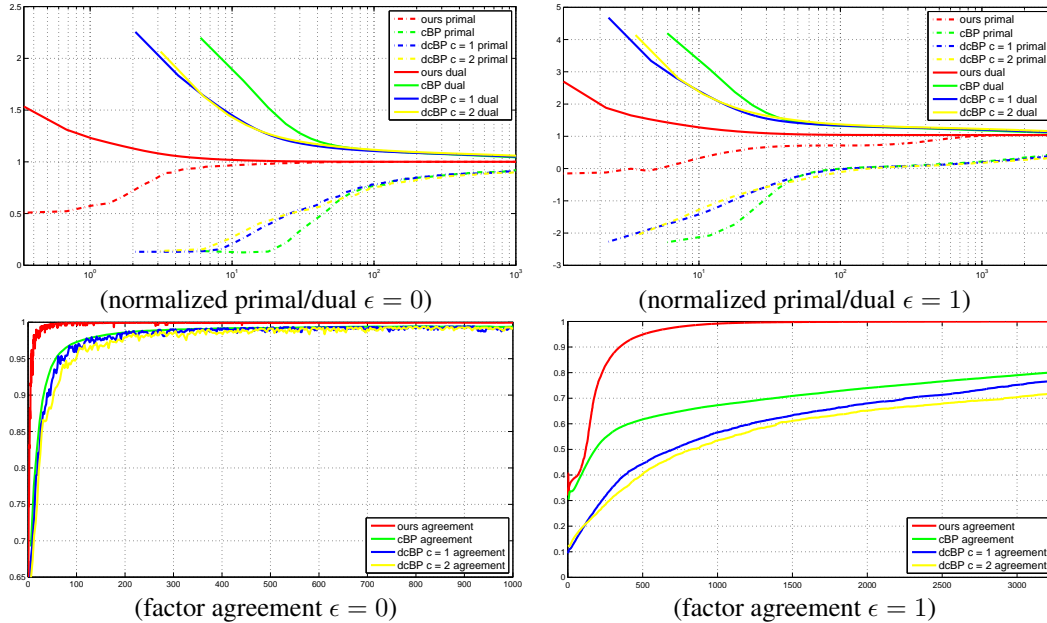

Figure 5: Average normalized primal/dual and factor agreement for $\epsilon = 1$ and $\epsilon = 0$.

potentials and pairwise regularization between neighboring superpixels. The third function, $\theta_{\text{comp}}$, couples the two tasks and encourages the layout and the segmentation to agree in their labels, *e.g.*, a superpixel on the left wall of the layout is more likely to be assigned the left-wall or the object label. The compatibility score decomposes into a sum of fifth-order scores, one for each superpixel, *i.e.*, $\theta_{\text{comp}}(s) = \sum_{i>4} \theta_{\text{comp},i}(s_1, \ldots, s_4, s_i)$. Using integral geometry [23], we can further decompose each superpixel score $\theta_{\text{comp},i}$ into a sum of third-order energies. As illustrated in Fig. 4(c), every superpixel variable $s_i$, $i > 4$ is therefore linked to 4-choose-2 third order functions of state-space size $6 \cdot 25^2$. These functions measure the overlap of each superpixel with a region specified by two layout ray angles $s_i, s_j$ with $i, j \in \{1, \ldots, 4\}$, $i \neq j$. This is illustrated in Fig. 4(b) for the area highlighted in purple and the blue region defined by $s_2$ and $s_3$. Since a typical image has around 250 superpixels, there are approximately 1000 third-order factors.

Following Claim 3 we recognize that the third-order functions are connected to at most two variables if we distribute the inference such that the layout task is assigned to one computer while the segmentation task is divided onto other machines. Importantly, this corresponds to a roughly equal split of the problem when using our approach, since all tasks are pairwise and the state-space of the layout task is higher than the one of the semantic-segmentation. Despite the third-order regions involved in the original model, every local inference task contains at most pairwise factors.

We use convex BP [35, 18, 9] and distributed convex BP [22] as baselines. For our method, we assign layout nodes to the first machine and segmentation nodes to the second one. Without introducing additional regions and pre-computations the workload of this split is highly unbalanced. This makes distributed convex BP even slower than convex BP since many messages are exchanged over the network. To be more fair to distributed convex BP, we split the nodes into two parts, each with 2 layout variables and half of the segmentation variables. For all experiments, we set $c_r = 1$ and evaluate the settings $\epsilon = 1$ and $\epsilon = 0$. For a fair comparison we employ a single core for our approach and convex BP and two cores for distributed convex BP. Note that our approach can be run in parallel to achieve even faster convergence.

We compare our method to the baselines using two metrics: **Normalized primal/dual** is a rescaled version of the original primal and dual normalized by the absolute value of the optimal score. This allow us to compare different images that might have fairly different energies. In case none of the algorithms converged we normalize all energies using the mean of the maximal primal and the minimum dual. The second metric is the **factor agreement**, which is defined as the proportion of factors that agree with the connected node marginals.

Fig. 5 depicts the normalized primal/dual as well as the factor agreement for $\epsilon = 0$ (*i.e.*, MAP) and $\epsilon = 1$ (*i.e.*, marginals). We observe that our proposed approach converges significantly faster

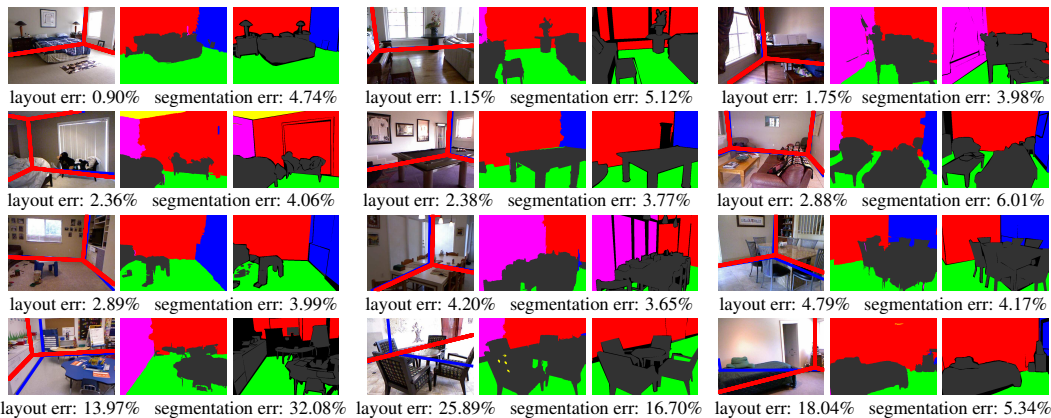

layout err: 0.90%  segmentation err: 4.74%  |  layout err: 1.15%  segmentation err: 5.12%  |  layout err: 1.75%  segmentation err: 3.98%

layout err: 2.36%  segmentation err: 4.06%  |  layout err: 2.38%  segmentation err: 3.77%  |  layout err: 2.88%  segmentation err: 6.01%

layout err: 2.89%  segmentation err: 3.99%  |  layout err: 4.20%  segmentation err: 3.65%  |  layout err: 4.79%  segmentation err: 4.17%

layout err: 13.97%  segmentation err: 32.08%  |  layout err: 25.89%  segmentation err: 16.70%  |  layout err: 18.04%  segmentation err: 5.34%

Figure 6: Qualitative Result ($\epsilon = 0$) : First column illustrates the inferred layout (blue) and layout ground truth (red). The second and third columns are estimated and ground truth segmentations respectively. Failure modes are shown in the last row. They are due to bad vanishing point estimation.

than the baselines. We additionally observe that for densely coupled tasks, the performance of dcBP degrades when exchanging messages every other iteration (yellow curves). Importantly, in our experiments we never observed any of the other approaches to converge when our approach did not converge. Tab. 1 depicts the time in seconds required to achieve a certain relative duality gap. We observe that our proposed approach outperforms all baselines by more than one order of magnitude. Fig. 6 shows qualitative results for $\epsilon = 0$. Note that our approach manages to accurately predict layouts and corresponding segmentations. Some failure cases are illustrated in the bottom row. They are largely due to failures in the vanishing point detection which our approach can not recover from.

## 5   Conclusions

We have proposed a partitioning strategy followed by a message passing algorithm which is able to speed-up significantly dual decomposition methods for parallel inference in Markov random fields with high-order terms and dense connections. We demonstrate the effectiveness of our approach on the task of joint layout and semantic segmentation estimation from single images, and show that our approach is orders of magnitude faster than existing methods. In the future, we plan to investigate the applicability of our approach to other scene understanding tasks.

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
