[Reviews · NeurIPS 2014]

Submitted by Assigned_Reviewer_26

Message Passing Inference for Large Scale Graphical
Models with High Order Potentials

The paper follows up on recent work on parallelizing message passing algorithms. The main contribution is dealing with higher order potentials. The main insight is that if large potentials are unary the message they send out is constant. By adding auxiliary factors this can be exploited (Fig 2 gives an example).

Clarity
The paper is clear and gives a nice overview of recent literature. A few small questions are outlined below.

Quality
The methods proposed appear to be sound.

Originality
The paper adds a twist to an earlier approach to parallelize message passing, namely to speed up models with high-order potentials. This addition does speed up results. Although it does require that the big potentials can be dealt with at least once.

Importance
The practical speed-up will be practical. The benefit is (authors please correct me if I have misunderstood in the rebuttal) to reduce the number of times the big potential is treated: from the number of iterations to 1. The example application in vision is interesting.

Questions:
- In the example in Fig 2 the big factor is in the overlap between machines. What impact does that have on the double loop algorithm? Is it just communication overhead?
- The messages are constant not just for unary potentials but any tree subgraph at the periphery. Could that be leveraged to reduce the communication overhead?
Summary: Summary: an interesting but perhaps modest in size extension to a promising parallelization of message passing. An interesting vision application is used to demonstrate the method.

Submitted by Assigned_Reviewer_41

The paper proposes a modification of the dual-decomposition approach for inference in graphical models that can handle high order potentials. It results a novel message passing algorithm adapted to efficient parallel inference.
The paper is clear and the improvement based on a trick adding appropriate regions in the region graph. I'm not truly able to assess the originality of the work compare to the existing literature which seems to be well cited in the paper. But the subject is of importance as being able to deal with large scale graphical models is useful. However it seems that applying the proposed technique on a specific subject requires to investigate its applicability, which seems to be non trivial.
Summary: The paper addresses an important task. It proposes an improvement of message passing techniques in term of computational efficiency. It is well positioned but not very novel in terms of methodology.

Submitted by Assigned_Reviewer_43

The paper improves on previous work on parallelizing MRF inference. The main insight is that when passing messages, summations over large state spaces can be delayed until the appropraite messages across computers are received. This is performed elegantly by adding new regions (typically correspodning to variable cliques)containing unions of high order parent regions' children, thus separating the higher order region for computation until the message from another computer arrives.

Results are impressive, far better than the baselines compared with. The paper was carefully written (perhaps overoptimized in some ways).

The two pieces of criticisms are:

1. The basic idea is relatively straightforward (but, has not been done before, apparently).
2. The writing is so dense and technical that it takes a while to get through the math and understand the relatively simple idea. Perhaps the authors should have a less abstract example (e.g., define an image task and a small number of regions, and then explain what the various messages mean). The details of the experiments (the expplanation/description of the task and the dataset) are also very compressed. Generally, the paper is succinct to the point that it has to be reread several times to be understood (but all details are there - that's why I say it was overoptimized).

THe above pieces of criticism are there just to help the authors in writing. I do, however, support acceptance (even in current form).
Summary: see above.
Author Feedback
Author rebuttal: We thank the reviewers for their comments. We will incorporate all their suggestions in a revised version.

To R26: The reviewer understood correctly that the construction reduces the number of accesses to high-order potentials. As message passing computation is mainly dominated by high-order factors, significant speed-ups are achieved by the proposed approach.

To R26: In the example of Fig. 2 the advantage of the proposed approach results from the fact that we partition the task on two machines. Having decomposed the problem, the `between-machine’ messages are constant during the machine-local updates. Convergence speed of the within-machine problems can then be further improved by introducing additional regions in a way that renders the high-order regions singly-connected. This property leaves the messages corresponding to high-order regions unchanged during machine-local iterations. As a consequence we can compute messages corresponding to high-order regions once upon having received a new `between-machine’ message. Improvements are very significant since those regions are high-order.

To R26: The reviewer is right in that the same scheduling technique could also be applied to tree subgraphs.

To R41: Obtaining efficient inference given a region-graph representation depends on the partitioning of the variables to machines and the newly introduced regions. The optimal partitioning is problem dependent. As a general rule, we should partition the variables within the scope of a high-order factor onto multiple machines while adding as little local regions as possible to obtain high-order terms that are locally singly-connected.

To R43: We will clarify the paper and try to reduce the level of over-optimization in the writing.